# Effects of Surface IR783 Density on the In Vivo Behavior and Imaging Performance of Liposomes

**DOI:** 10.3390/pharmaceutics16060744

**Published:** 2024-05-30

**Authors:** Qianqian Long, Xinmin Zhao, Lili Gao, Mengyuan Liu, Feng Pan, Xihui Gao, Changyou Zhan, Yang Chen, Jialei Wang, Jun Qian

**Affiliations:** 1School of Pharmacy, Department of Thoracic Medical Oncology, Fudan University Shanghai Cancer Center, Shanghai Medical College, Fudan University, Shanghai 200032, China; longqianqian815@sjtu.edu.cn (Q.L.); mizuyi@hotmail.com (X.Z.); 23211030016@m.fudan.edu.cn (M.L.); 21211030025@m.fudan.edu.cn (F.P.); 2Department of Pathology, Pudong New Area People’s Hospital, Shanghai 201299, China; gaolili@shpdph.com; 3School of Basic Medical Sciences, Fudan University, Shanghai 200032, China; gaoxihui@fudan.edu.cn (X.G.); cyzhan@fudan.edu.cn (C.Z.); 4Key Laboratory of Separation Science for Analytical Chemistry, Dalian Institute of Chemical Physics, Chinese Academy of Sciences, Dalian 116023, China; chernyang1987@gmail.com

**Keywords:** IR783, liposome, modification density, tumor imaging, protein corona

## Abstract

**Background:** Nanoparticles conjugated with fluorescent probes have versatile applications, serving not only for targeted fluorescent imaging but also for evaluating the in vivo profiles of designed nanoparticles. However, the relationship between fluorophore density and nanoparticle behavior remains unexplored. **Methods:** The IR783-modified liposomes (IR783-sLip) were prepared through a modified ethanol injection and extrusion method. The cellular uptake efficiency of IR783-sLip was characterized by flow cytometry and fluorescence microscope imaging. The effects of IR783 density on liposomal in vivo behavior were investigated by pharmacokinetic studies, biodistribution studies, and in vivo imaging. The constitution of protein corona was analyzed by the Western blot assay. **Results:** Dense IR783 modification improved cellular uptake of liposomes in vitro but hindered their blood retention and tumor imaging performance in vivo. We found a correlation between IR783 density and protein corona absorption, particularly IgM, which significantly impacted the liposome performance. Meanwhile, we observed that increasing IR783 density did not consistently improve the effectiveness of tumor imaging. **Conclusions:** Increasing the density of modified IR783 on liposomes is not always beneficial for tumor near-infrared (NIR) imaging yield. It is not advisable to prematurely evaluate novel nanomaterials through fluorescence dye conjugation without carefully optimizing the density of the modifications.

## 1. Introduction

In the past few decades, cancer has become one of the main causes of death worldwide [1]. The major directions to combat cancer are early diagnosis and precision treatment. In this regard, functionalized nanomaterials, due to their targeting potential, have been widely studied in the fields of cancer detection and targeted therapy [2,3]. For most functional nanoparticles, the surfaces are usually coated with polyethylene glycol (PEG), which is capable of slipping away from the reticuloendothelial system, thereby prolonging blood retention [4,5]. Furthermore, functional ligands that possess tumor-targeting moieties are usually further coupled with PEG, which can accurately bind to tumor-specific receptors and antigens, a process referred to as active targeting [6,7]. Consequently, PEG–ligand conjugates are broadly used to decorate multifarious nanocarriers [8], such as liposomes [9], micelles [10], nanoemulsions [11], and inorganic hybrid nanoparticles [12].

One of the prevalent applications of modifiable nanomaterials is to serve as an attractive and easily assembled platform for imaging modules [13,14], which play pivotal roles in various imaging techniques. Near-infrared (650–900 nm, NIR) fluorescence imaging, based on NIR fluorophore probes, featuring a combination of tissue-penetrated signals and minimally interfering backgrounds, are commonly deemed as ideal candidates for live imaging [15,16]. A variety of imaging nanoprobes modified with NIR fluorophores have been developed [13]. For instance, cy5.5 can be linked to the surface of mesoporous organosilica nanoparticles (MONs) [17], magnetic nanoparticles [18], and polymeric nanoparticles [19]. It is noteworthy that other functional targeting ligands, such as RGD [17], U11 peptide [18], and folic acid [19], are usually co-decorated on these nanoparticles due to the lack of tumor-targeting capability of the dyes themselves. However, several NIR dyes, including IR-783 and IR-808 (MHI-148), have been proven to have preferential uptake in human cancer cell lines, including lung, breast, and prostate cancer cells, as well as preferential accumulation in human tumor xenografts [20], which are mediated by organic-anion-transporting polypeptides on the cell membrane of tumor cells [21]. Thus, these dyes have the potential to exert a dual role in tumor targeting, as well as NIR imaging. Based on this, it may be simple to couple them on the surface of nanomaterials to achieve targeted imaging. To the best of our knowledge, there are few reports on this strategy available in the literature.

It is accepted that surface engineering of nanomaterials dominates the in vivo performance of the complexes, including pharmacokinetics, biodistribution, targeting efficiency, and consequent safety [22,23,24,25]. So far, several studies have shown that the density of ligands conjugated on the nanocarrier surface affects the targeting efficiency of nanoparticles through modulation of physicochemical properties and/or the affinity between ligands and receptors. The density of ligands linked to the nanovesicles usually increases the cellular uptake efficiency, but it is not constantly in line with the tumor-targeting efficiency in vivo [26,27,28,29,30,31,32,33]. It is worth pointing out that the ligands in these reports are mostly peptides, proteins, and small molecular receptor-specific ligands, such as folic acid, methotrexate, and anisamide, rather than dyes [8]. To the best of our knowledge, the in vivo effects of the density of fluorophores conjugated to nanocarriers have barely been investigated. Moreover, whether this factor affects the imaging performance of the nanoprobes remains unrevealed, warranting further verification and analysis.

In this work, considering the prominent tumor-targeting ability of IR783 [34], we first developed IR783-modified liposomes (IR783-sLip) as a representative imaging probe by virtue of a conjugate of IR783 and PEG2000-DSPE, and we chose lung cancer as a tumor model due to its high incidence and mortality. Next, we investigated the in vitro characteristics and in vivo performance of four IR783-sLip with different IR783 densities. As it turned out, the highest IR783 density rendered liposome with the most efficient cellular uptake and intensive fluorescence in tumors, which did not translate into the most superior tumor-targeting imaging performance. Moreover, the density was inversely correlated with the pharmacokinetic performance. This unexpected finding could be attributed to the natural IgM in the protein corona, a dynamic coating that forms around nanomaterials when they are exposed to biological mediums.

These findings will shed light on other engineered nanoparticles decorated with fluorescent dyes or other imaging agents, and could partly explain the discrepancies observed between basic research and clinical applications involved in the evaluation of in vivo behaviors of nanomedicines via the strategy of conjugating imaging agents onto their surfaces.

## 2. Materials and Methods

### 2.1. Materials

Hydrogenated soy phosphatidylcholine (HSPC), cholesterol, N-(Carbonyl-methoxypolyethylene glycol 2000)-1,2-distearoyl-sn-glycerol-3-phosphoethanolamine (mPEG2000-DSPE), and 1,2-distearoyl-sn-glycero-3-phosphoethanolamine-N-[amino(polyethyleneglycol)-2000] (DSPE-PEG2000-NH2) were purchased from A.V.T. Biomed Co., Ltd. (Shanghai, China). COOH- IR783 was gifted by the School of Chemistry and Chemical Engineering, Shanghai Jiao Tong University. The 2-(7-Azabenzotriazol-1-yl)-N,N,N′,N′-tetramethyluronium hexafluorophosphate (HATU), N,N-diisopropylethylamine (DIPEA), and Dimethyl sulfoxide (DMSO) were from Sinopharm Chemical Reagent Co., Ltd., Shanghai, China. The 1,1′-dioctadecyl-3,3,3′,3′-tetramethylindocarbocyanine perchlorate (DiI) was from Sigma-Aldrich, St. Louis, MO, USA. The 5-carboxyfluorescein (FAM) was from Aladdin–Holdings Group Co., Ltd., Shanghai, China. Sephadex G50 column was from Sigma-Aldrich, St. Louis, MO, USA. Dulbecco’s Modified Eagle’s medium (DMEM) was from Gibco, Shanghai, China. FBS was from Gibco, Carlsbad, CA, USA. The Cell Counting Kit-8 (CCK-8) was from GLPBIO, Shanghai, China. Hoechst 33342 was from ThermoFisher Scientific Inc., Shanghai, China. Matrigel (356234) was from Corning BD Biocoat, Bedford, MA, USA. The Fast Silver Stain Kit (P0017S) was from Beyotime lnc., Shanghai, China. The Goat Anti-Mouse IgM (HRP) Antibody (ab97230) was from Abcam, Shanghai, China. The Clarity Western ECL substrate was from Bio-Rad, Shanghai, China.

The A549 cell line and human umbilical vein endothelial cell line (HUVEC) cell line were purchased from the Type Culture Collection of the Chinese Academy of Sciences (Shanghai, China) and were both cultured in Dulbecco’s Modified Eagle’s medium (DMEM) containing 10% fetal bovine serum (FBS) at 37 °C, under a humidified atmosphere containing 5% CO_2_. BALB/c and BALB/c nude mice were purchased from Shanghai LingChang Biotech Co., Ltd. (Shanghai, China) and housed in a specific pathogen-free (SPF) animal facility.

### 2.2. Synthesis of IR783-PEG2000-DSPE

IR783-PEG2000-DSPE was synthesized by condensation reaction. In brief, 54.2 mg of COOH-IR783 and 45.6 mg of HATU were dissolved in 500 μL of DMSO solution, and the mixture was then added to 3.5 mL of DMSO containing 110 mg of NH2-PEG2000-DSPE and 55.8 μL of DIPEA, followed by reaction for an hour with vigorous stirring at room temperature (RT). The residual COOH-IR783 and salt were removed by dialysis against distilled water (MWCO 3000 Da) for 72 h. The IR783-PEG2000-DSPE product was finally obtained by freeze-drying.

### 2.3. Preparation and Characterization of Liposomes (sLip, sLip/DiI, IR783-sLip, sLip/FAM, and IR783-sLip/FAM)

Liposomes were prepared via a modified ethanol injection [35] and extrusion method [36]. To prepare PEGylated blank liposomes (sLip), briefly, a mixture of HSPC/cholesterol/mPEG2000-DSPE (52/43/5 in molar ratio) [37] was dissolved in ethanol at 65 °C while stirring to evaporate most of the solvent. Preheated sterile saline at 65 °C was then slowly dispensed in the mixture (HSPC at 10 mM). The mixture was stirred for 20 min, followed by sequential extrusion through polycarbonate membranes with pore diameters of 200 nm and 100 nm. For liposome loading DiI (sLip/DiI) preparation, DiI was dissolved in ethanol along with the lipid material (0.5%), and consequent operations were conducted as described above. IR783-conjugated liposomes (1% IR783-sLip, 2% IR783-sLip, 3.5% IR783-sLip, and 5% IR783-sLip) were prepared following the same method except for the addition of 1%, 2%, 3.5%, and 5% IR783-PEG2000-DSPE and reduction of mPEG2000-DSPE to 4%, 3%, 1.5%, and 0%, respectively. FAM-loaded sLip (sLip/FAM) and FAM-loaded IR783-sLip (IR783-sLip/FAM) were prepared based on the same approach. Specifically, FAM was dissolved in sterile saline (65 °C, pH 7.0) at first, followed by the same procedure as outlined above, with the final removal of free FAM using a Sephadex G50 column [38]. The FAM concentration was measured under conditions of excitation (Ex) at 494 nm and emission (Em) at 580 nm using a Spark Cyto automatic microplate reader (Tecan Group Ltd., Männedorf, Zurich, Switzerland). The size/stability and zeta potential (electrokinetic potential in colloidal dispersions) of liposomes were measured in double-distilled water using a Zetasizer Nano ZS90 analyzer (Malvern Instrument, Southborough, MA, USA).

### 2.4. Characterization of Protein Corona In Vitro and In Vivo

To characterize protein corona in vitro, whole blood samples were collected from untreated BALB/c mice. The collected blood was allowed to stand for 30 min (RT), followed by centrifugation at 1000× *g* for 10 min, and mouse serum was obtained. One hundred microliters of serum was incubated with isovolumetric liposomes in low-protein-binding tubes at 37 °C for 1 h, then 800 μL of chilled phosphate-buffered saline (PBS) was added, mixed, and centrifuged at 14,000× *g* for 30 min (*n* = 3). The sediments were rinsed with 300 μL of chilled PBS three times, as described above, and finally suspended in a mixture of 30 μL of PBS with an additional 7.5 μL of sample loading buffer (5×, reducing). Protein pellets were separated by molecular weight on a 4–20% gradient polyacrylamide gel and stained using the Fast Silver Stain Kit. The protein band at 72 kD was verified via the Western blot assay. For characterization of protein corona in vivo, whole blood samples were collected from BALB/c mice and intravenously injected with sLip/DiI, and 1%, 2%, 3.5%, and 5% IR783-sLip (39.25 mg of HSPC/kg of a mouse) at 1 h post-injection. The subsequent procedures were conducted as described above.

### 2.5. Western Blot Analysis

Protein pellets were separated by molecular weight on an 8% gradient polyacrylamide gel and transferred to PVDF membranes. TBST (TBS plus 0.1% Tween 20) containing 5% skimmed milk was used to block the nonspecific sites on the PVDF membrane for 1 h (RT). The PVDF membranes were rinsed with TBST 3 times to remove the extra blocking buffer and incubated with goat anti-mouse IgM antibody conjugated with HRP (1:2000 dilution; RT, 1 h). After removing the uncombined antibody by rinsing three times with TBST, the bound antibody was visualized by the Clarity Western ECL substrate and imaged (ChemiScope 6000, Clinx Co., Ltd., Shanghai, China). Data were analyzed using Image J 1.35 k software.

### 2.6. Cytotoxicity Analysis

The cytotoxic effects of sLip and IR783-sLip with various densities of IR783 modification were evaluated by the CCK-8 assay. HUVEC were seeded in 96-well plates at a density of 2 × 10^3^ cells/well. After 24 h, the cells were treated with sLip, 1% IR783-sLip, 2% IR783-sLip, 3.5% IR783-sLip, and 5% IR783-sLip (concentrations of HSPC in each liposome varied from 2.5 mg/mL to 0.001 mg/mL), and liposome-free culture media as the control, followed by incubation for 72 h. Thereafter, the medium was substituted with a medium containing 10% CCK-8 in 100 μL and incubated at 37 °C for 1 h. The absorbance of each well was read at 450 nm using an automatic microplate reader (Tecan Group Ltd., Männedorf, Zurich, Switzerland).

### 2.7. Cellular Uptake

The cellular uptake efficiency of IR783-sLip with various ratios of IR783 modification by A549 cells was characterized by flow cytometry and fluorescence microscope imaging. A549 cells (5 × 10^4^ per well) were seeded into a 24-well plate. After 12 h, cells were subsequently incubated with sLip/FAM, and 1%, 2%, 3.5%, and 5% IR783-sLip/FAM (HSPC at 1 mM) at 37 °C for 8 h in culture medium. Cells were then rinsed 3 times with PBS, digested with trypsin, and collected via centrifugation. Cells were resuspended in 400 μL of PBS and their fluorescence intensities were captured and quantified by a FACS Aria II flow cytometer (BD Biosciences, San Jose, CA, USA). The FITC signals within the range of 10^0^–10^5^ were gated and calculated.

To visualize the uptake of IR783-sLip by A549 cells, fluorescence microscope imaging was also conducted. Cells (1.5 × 10^5^ per well) were seeded into a 12-well plate. The consequent culture and incubation procedures were carried out as previously described. Then, the cells were washed 3 times with PBS and fixed with 4% formaldehyde for 15 min (RT). Next, the cell nuclei were stained by Hoechst 33342 for 10 min (RT). Finally, cells were observed through a fluorescence microscope (MF53-N, Mshot, Optoelectronic Technology Co., Ltd., Guangzhou, China).

### 2.8. Pharmacokinetic Evaluation

The 6–8-week-old male BALB/c mice were randomly allocated to 5 groups (*n* = 3–4) and administered with sLip/DiI or sLip-IR783 (1%, 2%, 3.5%, or 5%) through the caudal vein (39.25 mg of HSPC per kg of a mouse). At predetermined time points (0.5 h, 1 h, 2 h, 4 h, 8 h, 12 h, and 24 h), blood was collected, and plasma was obtained by centrifugation at 1000× *g* for 10 min. The concentration of liposomes in plasma was detected by an automatic microplate reader (Ex at 545 nm and Em at 580 nm for sLip/DiI; Ex at 740 nm and Em at 780 nm for IR783-sLip).

### 2.9. Subcutaneous Tumor Model

The 6–8-week-old male BALB/c nude mice were randomized into 4 groups (*n* = 3–4). A549 cells were suspended in a mixture of PBS and Matrigel (1:1; 1 × 10^8^ cells/mL) and subcutaneously injected into the right flank of the mice (200 μL/mouse). The tumor growth was monitored every 3 days [37] post-injection. When the tumor diameter reached 8–10 mm, the experiments were carried out on the animals.

### 2.10. In Vivo and Ex Vivo Fluorescence Imaging

Tumor-bearing BALB/c nude mice (*n* = 3) were administered 1% IR783-sLip, 2% IR783-sLip, 3.5% IR783-sLip, and 5% IR783-sLip (39.25 mg of HSPC/kg per mouse) through the vein tail, respectively, and imaged using the IVIS Lumina II fluorescence imaging system (Caliper Life Sciences, Hopkinton, MA, USA) at specific time points (1 h, 2 h, 4 h, 6 h, 8 h, 12 h, and 24 h after drug administration). To further characterize the biodistribution of the aforementioned IR783-sLip, the major organs/tissue (heart, liver, spleen, lung, and kidney) and tumors of mice were dissected promptly at the 24 h time point, followed by ex vivo imaging with the same fluorescence imaging system (Ex at 745 nm and Em at 800 nm). The fluorescence signals were analyzed using Living Image software 4.2. The TBR (tumor-to-background ratio) was calculated as the ratio of the average fluorescence intensity of the tumor ROI (region of interest) against an average of three equally sized background ROI regions, which were contralateral, caudal, and cephalic.

### 2.11. In Vivo Safety Evaluation

The liver and kidney functions of mice were measured with an automatic biochemical analyzer (Chemray 240, Shenzhen Leidu Biotech. Inc., Shenzhen, China). After administration with sLip and IR783-sLip (1%, 2%, 3.5%, and 5%) once every other day for four treatments, serum was collected for detection of alanine aminotransferase (ALT), aspartate aminotransferase (AST), total bilirubin (TBIL), and blood creatinine (CRE). In addition, H&E staining of the major organs (heart, liver, spleen, lung, and kidney) was also conducted. After being fixed in 4% paraformaldehyde solution for 24 h, the organs/tissue were dehydrated in graded ethanol, then embedded in paraffin, and cut into slices (5 μm), which were successively deparaffinized with xylene and then immersed in gradient ethanol to remove residual xylene. This was followed by rehydration with distilled water. Lastly, the slices were stained with hematoxylin and eosin (H&E). The histological characteristics of the organ/tissue were observed using an optical microscope (ICC50HD, Leica, Heidelberg, Germany).

### 2.12. Statistical Analysis

Data are presented as means ± standard deviations (SDs) and were analyzed with GraphPad Prism software 8.0.1. *p* < 0.05 was considered statistically significant (ns, no significant difference, *p* > 0.05; * *p* < 0.05, ** *p* < 0.01, and *** *p* < 0.001).

## 3. Results

### 3.1. Characterization of IR783-sLip

First, IR783-PEG2000-DSPE was chemically synthesized through the conjugation of COOH-IR783 and NH2-PEG2000-DSPE (Figure 1A). Stealth liposome (sLip, with a 5% molar ratio of mPEG2000-DSPE) and IR783-modified liposomes (1%, 2%, 3.5%, and 5% IR783-sLip, composed of a specific molar ratio of IR783-PEG2000-DSPE and mPEG2000-DSPE, in which PEG2000-DSPE accounted for 5% in total) were prepared through the modified ethanol injection and extrusion method (Figure 1B). Using dynamic light scattering, all formulations were estimated to be smaller than 150 nm, without significant size differences. IR783 modification was shown to have no significant influence on the zeta potential of liposomes (Appendix A). In addition, the liposomes were stable and had no obvious change in particle size for up to 30 days when stored at 4 °C.

### 3.2. Dense IR783 Modification Enhances the Liposomal Cellular Uptake Efficiency

Initially, to study the effects of IR783 density on cellular uptake efficiency, sLip/FAM (FAM-labeled sLip) and 1%, 2%, 3.5%, and 5% IR783-sLip/FAM (FAM-labeled IR783-sLip) were, respectively, incubated with A549 cells for 8 h. Here, FAM was encapsulated in the inner aqueous core of the liposomes for visualization of liposomes without excess modification on the liposomal surface as a confounding factor. As was observed through fluorescence microscopy imaging (Figure 2A), cells treated with IR783-sLip showed brighter green fluorescence representing FAM than sLip, and IR783-sLip with a higher IR783 density delivered a brighter fluorescence. Further examination by flow cytometry exhibited similar results. Compared to the sLip/FAM group, the IR783-sLip/FAM groups exhibited a much stronger fluorescence intensity. Moreover, the 5% IR783-sLip/FAM group presented the strongest fluorescence intensity, 5.9, 3.7, 3.1, and 1.3 times stronger than that of the sLip/FAM, and 1%, 2%, and 3.5% IR783-sLip/FAM groups, respectively. Meanwhile, it was noticed that the 1% and 2% IR783-sLip/FAM groups showed no significant differences (Figure 2B,C). These results demonstrated that the increase in the IR783 modification density could improve the cellular uptake of IR783-sLip.

### 3.3. IR783 Modification Impacts the Liposomal Pharmacokinetic Profile

Based on the results in vitro, we continued to investigate the role of the density of conjugated IR783 in liposomal in vivo behaviors, and pharmacokinetic studies were conducted first. BALB/c mice (*n* = 3–4) were intravenously administered sLip/DiI (DiI labeled sLip) or sLip-IR783 (1%, 2%, 3.5%, and 5%) with the same dose of HSPC (39.25 mg of HSPC per kg of mouse). At predetermined time points (0.5 h, 1 h, 2 h, 4 h, 8 h, 12 h, and 24 h), blood was collected, and the concentration of liposomes in plasma was determined by quantitation of fluorescence intensity. All IR783-conjugated liposomes were eliminated from plasma more rapidly than sLip/DiI. At 12 h post-injection, all IR783-sLip were almost undetectable in plasma, while 31.9% sLip/DiI remained (Figure 3A). In sharp contrast, the area under the curve (AUC) of 1%, 2%, 3.5%, and 5% IR783-sLip, dramatically decreased by 4.1-, 5.5-, 16.4-, and 41.2-fold, respectively. Moreover, as the IR783 density escalated from 1% or 2% to 3.5%, the AUC showed a statistically significant reduction, while the increased ratio from 3.5% to 5% showed no statistically significant consequence to AUC (Figure 3B). These results indicated that greater IR783 modification generally played an adverse role in liposomal retention in blood.

### 3.4. IR783 Modification Density Affects the Tumor-to-Background Ratio

Considering the results above, we further studied the living NIR imaging efficiency mediated by four IR783-sLip. As the timeline shows in Figure 3A, 1%, 2%, 3.5%, and 5% IR783-sLip were intravenously injected into mouse models bearing the subcutaneous lung cancer tumor. The mice were imaged in vivo at predetermined time points (1 h, 2 h, 4 h, 6 h, 8 h,12 h, and 24 h) post-injection. All groups showed a higher tumor-to-background ratio (TBR; for a definition, see the Methods Section) at the 8 h time point than any other time point (Appendix A and Figure 3D). By statistical comparison, the average fluorescence intensity of all time points in the tumor region of interest (ROI) of the 5% IR783-sLip group was the strongest, which was 1.97-, 1.88-, and 1.30-fold of the 1%, 2%, and 3.5% IR783-sLip groups, respectively. Additionally, the 3.5% IR783-sLip group exhibited signals inferior to 5% IR783-sLip and superior to 1% and 2% IR783-sLip groups, and the latter two groups showed no significant differences (Figure 3E). Both 3.5% and 5% IR783-sLip groups had the highest TBR during successive imaging, while the 1% and 2% IR783-sLip groups were both the lowest in TBR (Figure 3F). These results indicated that the increase in density of IR783-modified liposomes, for example from 1% or 2% to 3.5%, could enhance the TBR. However, the increase in IR783 density from 3.5% to 5% favored a brighter fluorescent image of tumor lesions, though it enhanced signals of other normal tissues as well. This compromised the highlighted tumor imaging, indicating that denser IR783 modification on liposomes was not invariably accompanied by a more selective tumor-targeting ability.

### 3.5. IR783 Modification Density Affects the Distribution of Liposomes

Further, ex vivo imaging of the major organs 24 h post-administration revealed that different IR783-sLip groups had different biodistribution modes. In all groups, the top four primarily distributed organs/tissues of fluorescence were the liver, tumor, spleen, and kidney. The distribution to the liver was the highest in all groups, while secondary distribution patterns varied among groups. Specifically, in terms of fluorescence distribution, in the 1% IR783-sLip group, the tumor, spleen, and kidney ranked second together, with no significant difference. In the 2% IR783-sLip group, the tumor ranked second, while the spleen and kidney ranked third. In the 3.5% IR783-sLip group, the tumor and spleen ranked second, while the kidney was comparable to the spleen but less than the tumor. In the 5% group, the spleen ranked second, while tumor and kidney ranked third together (Figure 4A,B). Moreover, a horizontal comparison of the fluorescent distribution modes among groups showed that the increase in the IR783 modification density was not always associated with the synchronized increase in fluorescence intensity in tumors and major organs. Specifically, when the IR783 density increased from 1% to 2%, the fluorescence accumulation in the liver and spleen did not change, while that in tumors increased significantly. When the density increased from 2% to 3.5%, the accumulation of fluorescence in the liver significantly increased, while that in the tumor and spleen did not show significant alteration. When the density increased from 3.5% to 5%, the fluorescence intensity in the liver and tumor both dramatically decreased instead, comparable to 1% IR783-sLip. Although the accumulation of fluorescence in the spleen did not show any further significant increase, it was higher than that in the tumor. There was no significant difference in the fluorescence distribution in kidneys among all groups (Figure 4C).

### 3.6. IR783 Modification Density Correlates with Protein Corona Absorption on Liposomes

As reported, the in vivo performance of liposomes was strongly linked with the plasma-derived “protein corona” absorbed onto its surface [18,19,20,21,22], a phenomenon in which natural IgM could play a dominant role [33]. Herein, the protein corona composition of IR783-sLip was specifically studied. As shown in Figure 5A, IR783-sLip was incubated with isometric serum for 1 h in vitro, followed by centrifugation, and the protein corona obtained was separated by SDS-PAGE and visualized by silver staining. The silver staining result demonstrated that protein corona, especially the region between 66 and 95 kD, increased along with the gradual increase in density of IR783-modified liposomes, which was thereafter ascertained as natural IgM (mu chain, 72 kD) by Western blot (Figure 5C). This suggests that denser IR783 modification causes more IgM deposition on IR783-sLip in vitro. To further investigate the interaction between IR783-sLip and IgM in vivo, we characterized the protein corona of different IR783-sLip groups using a similar method, utilizing the serum collected after administration of IR783-sLip. The silver staining result was similar to the pattern of protein corona in vitro (Figure 5B). Further semi-quantitative Western blot results showed (Figure 5C,D) that the correlation of the IR783 modification density and natural IgM absorption was found consistent with that observed in vitro. Compared to sLip, 1% IR783-sLip absorbed dramatically more IgM. When the IR783 density rose from 1% to 2%, there was no significant increase in IgM absorption. However, the increase in IR783 density, from 2% to 3.5% and then to 5%, significantly augmented IgM deposition.

Taking into consideration that more absorption of IgM could lead to rapid clearance of some liposomes, we further conducted the curve fit between normalized natural IgM and the AUC data (Figure 5E), and the trend was found to be consistent with the existing reports involving other liposomes. These results indicate that natural IgM is also a potential predictor of the pharmacokinetic performance of IR783-sLip.

### 3.7. Safety Evaluation of IR783-sLip

Finally, as a novel nanomaterial, the safety of IR783-sLip was evaluated both in vitro and in vivo. On the one hand, after co-incubation with human umbilical vein endothelial cells for 72 h, the relative cell viability curves in each group showed a consistent trend in fluctuations: neither the IC_50_ value of any IR783-sLip nor of sLip was reached (Appendix A), indicating that IR783-sLip was nontoxic and the differences in IR783 modification density exerted no effect on the safety of IR783-sLip in vitro. On the other hand, after administration once every other day for four treatments, liver and kidney functions presented by biochemical indices as well as histopathological manifestations of the major organs of mice were evaluated. The results showed that neither IR783-sLip nor sLip induced any kidney/liver abnormality (Figure 6A–D) and no perceptible toxicity was exhibited in HE-stained sections (Figure 6E).

## 4. Discussion

It was reported that NIR photo-diagnostic agents, such as IR780, IR783, and IR825, can be loaded into liposomes to serve as stable nanotheranostic agents for improved photothermal/photodynamic therapy of tumors [39,40,41]. These fluorescent agents were encapsulated either in the liposomal hydrophilic core or within the lipidic bilayer of liposomes, entailing liposomal surface modification for active tumor-targeting ability. Here, IR783 was directly conjugated with the surface lipid, rather than encapsulated into the inner cavity of liposomes, and we expected a dual role of active tumor-targeting and in vivo imaging. However, whether the in vivo behaviors and imaging performance of liposomes conjugated with fluorescent groups are related to the density of the fluorophore remains unknown.

We took IR783-sLip as an example to explore whether the density of surface IR783 had an impact on the pharmacokinetic profile, tissue distribution, imaging performance, and protein corona composition of liposomes. Collectively, we found that the IR783 density did have an influence on the in vivo fate of the liposomes. Here, IR783 was directly conjugated with the surface lipid, rather than encapsulated into the inner cavity of liposomes, and hence played a dual role of active tumor-targeting and in vivo imaging.

Initially, we found that the presence of IR783 enhanced the cellular uptake efficiency of primitive liposomes, and the higher the density of linked IR783, the higher the liposomal uptake efficiency. In vivo, densely modified IR783 impaired the pharmacokinetic performance. Nonetheless, live imaging results showed that the NIR fluorescence intensity in the tumor ROI was consistent with the affinity of IR783-sLip to tumor cells in vitro. These results suggest that even if the liposomes with higher IR783 density were quickly eliminated from plasma, intensified fluorescence could still be exhibited in the tumor sites, even though the imaging performance was not excellent. Specifically, when the IR783 density increased from 3.5% to 5%, the fluorescence intensity in the tumor ROI was enhanced, while the TBR, surprisingly, did not show improvement, indicating that the background fluorescence of 5% IR783-sLip was considerably increased as well.

Among the four IR783-sLip groups studied, 5% IR783-sLip had a fluorescence intensity in the tumor ROI comparable to that in 2% and 3.5% IR783-sLip groups, but significantly stronger intensity than that of 1% IR783-sLip (5% IR783-sLip vs. 1% IR783-sLip, *p* < 0.01). The ex vivo imaging showed that 5% IR783-sLip, as well as 1% IR783-sLip, had the lowest fluorescence intensity in dissected tumors. These conflicting results suggested that the fluorescence in the tumor ROI during in vivo imaging mediated by 5% IR783-sLip did not equivalently present the 5% IR783-sLip itself, implying the existence of considerable fluorescent noise, which probably explains the limited TBR of 5% IR783-sLip. Therefore, although 5% IR783-sLip had the strongest fluorescence in the tumor ROI and the highest TBR along with 3.5% IR783-sLip, the tumor-selective imaging performance was inferior to 3.5% IR783-sLip.

Interestingly, we found that IR783 density severely impacted the composition of the protein corona absorbed by IR783-sLip, especially natural IgM, which was further found to be negatively correlated with the AUC. A recent study reported that high IgM absorption could lead to a series of alterations in the in vivo performance of liposomes modified with folic acid, Angiopep-2 peptide, and DCDX peptide, including their accelerated elimination from plasma. The high deposition of IgM could be attenuated by lowering the degree of modification [33]. Our observations involving this aspect were similar to the reported findings, although the modified fluorophore taking on the role of the tumor-targeting ligand in our study was an imaging agent rather than peptides or small molecular ligands. Additionally, Wang et al. identified that the complement activated by absorbed IgM on liposomes modified with folic acid (FA-sLip) could redirect the uptake of FA-sLip by the resident macrophages in the liver and spleen [37]. In our study, 5% IR783-sLip presented a much heavier accumulation in the spleen and liver than that in the tumor, which may lead to a discouraging tumor-targeted imaging performance. These findings indicate that the effects of IR783 density exerted on the liposomal in vivo behaviors were probably indirectly mediated by the IgM.

Additionally, when IR783 density increased from 1% to 2%, the cellular uptake efficiency, AUC, IgM content, fluorescence intensity of the tumor ROI, and the TBR were not significantly altered. This indicates that there was a plateau in the trend of the in vivo performance of liposomes with the varying IR783 density. All the findings clarified that an increase in the surface IR783 density may not play a positive role in liposomal performance in vivo and may even contradict our initial assumptions.

Overall, we demonstrated that surface IR783 density plays a crucial role in the pharmacokinetics, biodistribution, and tumor NIR imaging of liposomes, where IgM in the absorbed protein corona could be a potential mediator. Considering the existing data, 3.5% IR783-sLip appears to be the most efficient nanoprobe for tumor NIR imaging. The time window of in vivo imaging is broad, ranging from 6 h to 24 h post-administration, which is convenient for the potential application of intraoperative imaging. Notably, 8 h post-administration is the most appropriate time point for the highest TBR. Our study provides a novel perspective on the elaborate design and development of fluorophore-conjugated nanoparticles. More importantly, our findings revealed a potential cause of the contradictory results observed between basic research and clinical applications involving similar functionalized materials. Based on our observation, it is not advisable to track and preclinically evaluate the nanoparticles in vivo through fluorescence imaging with the imaging agents linked to nanovesicles if the density of the modified dyes is not already weighed and optimized.

Despite the investigation and findings in this study, there are some limitations that should be acknowledged. Firstly, IR783-modified liposomes were bio-distributed in the liver the most, a common disadvantage of all liposomes, which may induce the limitations of IR783-modified liposomes to target orthotopic liver cancer or cancer on organs adjacent to the liver. Therefore, it is more helpful to apply it in intraoperative imaging for minimal residual disease than early systemic screening for cancer or combine it with other diagnostic approaches. Moreover, this can be addressed by general strategies, for example, blocking the reticuloendothelial system reversibly and temporarily. Secondly, the role that natural IgM plays in the pathway to the in vivo behaviors of fluorophore-conjugated nanoparticles is not very clear. Further work is required to address this issue.

## 5. Conclusions

The surface density of IR783 on liposomes played a crucial role in determining their in vivo fate. Contrary to expectations, increasing the density of modified IR783 on liposomes did not always result in a beneficial effect on tumor NIR imaging. This finding underscores the need for careful optimization of the modification density of fluorescence dyes on nanomaterials for preclinical evaluation of their in vivo profiles.

Our study revealed that the liposomes with the highest IR783 density did not necessarily exhibit the most optimal tumor NIR imaging performance. The relationship between IR783 density and tumor imaging performance is complex, with other factors, such as natural IgM absorbed by liposomes, that potentially mediate this effect. As a result, blindly conjugating fluorescence dyes to nanomaterials without considering the optimized modification density is not recommended for preclinical evaluation of in vivo behaviors.

These findings highlighted the importance of meticulous optimization of modification densities of imaging agents on nanomaterials to achieve desirable in vivo performance. Careful consideration and tailoring of the modification density of fluorescence dyes or other imaging agents are necessary to avoid potential discrepancies between basic research and clinical applications. Further research is warranted to elucidate the underlying mechanisms and develop robust strategies for the design and evaluation of nanomaterials with imaging capabilities in cancer diagnosis and therapy.

## Figures and Tables

**Figure 1 pharmaceutics-16-00744-f001:**
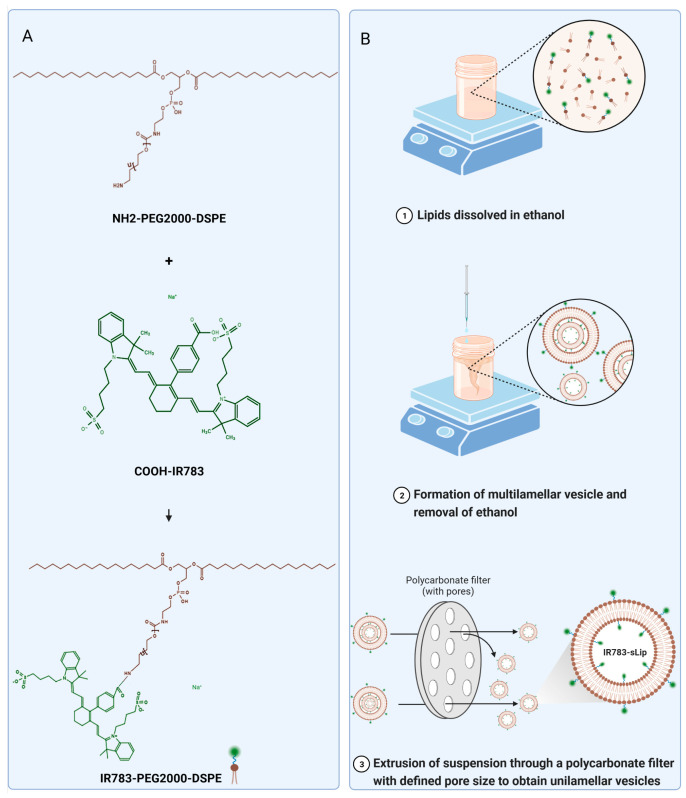
Illustration of the preparation of IR783-modified liposomes. (**A**) Synthesis of IR783-PEG2000-DSPE. (**B**) Preparation of IR783-modified liposomes (IR783-sLip) via the modified ethanol injection and extrusion method.

**Figure 2 pharmaceutics-16-00744-f002:**
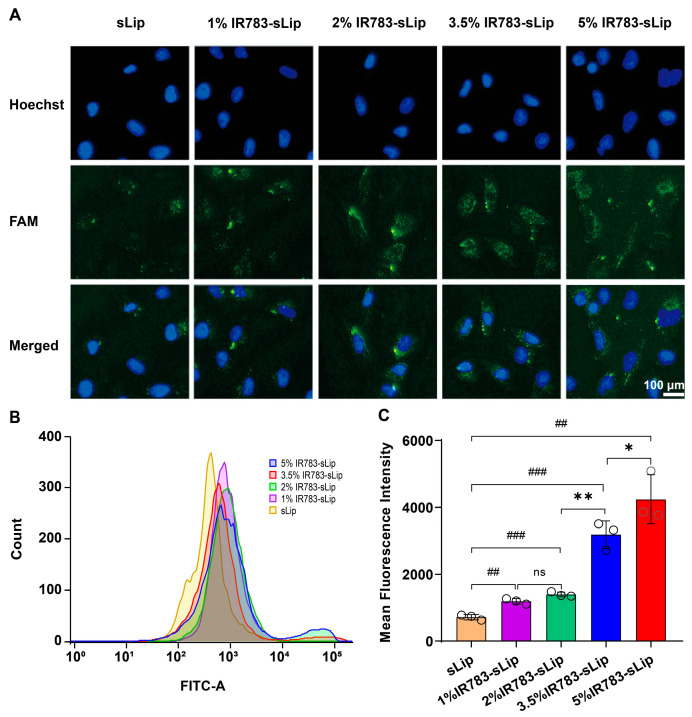
Effect of the density of conjugated IR783 on the cellular uptake efficiency of liposomes. A549 cells were incubated with 1%, 2%, 3.5%, and 5% IR783-sLip/FAM and sLip/FAM for 8 h, respectively. (**A**) Representative fluorescence microscopy image of the cells. Green shows FAM, blue shows Hoechst 33342 staining of the cell nuclei. (**B**) Flow cytometry histograms showing intracellular fluorescence. (**C**) Detected fluorescence intensities, normalized to A549 cells as controls. The bars represent mean ± SD (*n* = 3). ## *p* < 0.01, ### *p* < 0.001 by Student’s *t*-test; ns, no significance; * *p* < 0.05; ** *p* < 0.01 using one-way ANOVA within all IR783-sLip groups.

**Figure 3 pharmaceutics-16-00744-f003:**
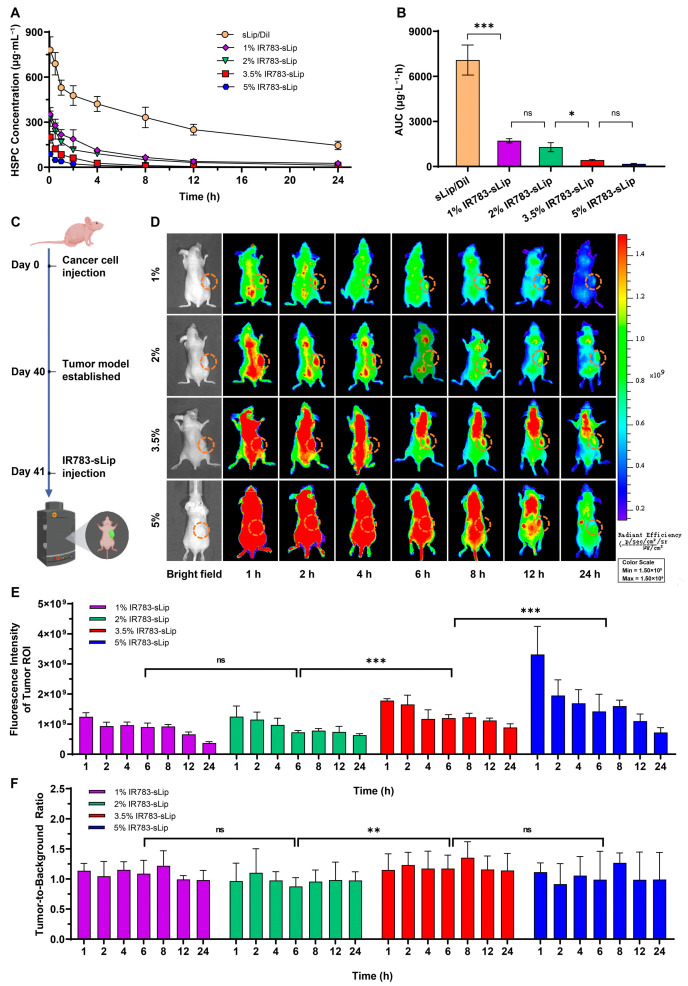
Effect of the density of conjugated IR783 on the liposomal pharmacokinetic profile and near-infrared (NIR) fluorescence live imaging performance of IR783-sLip. BALB/c mice were intravenously dosed with 1%, 2%, 3.5%, and 5% IR783-sLip. (**A**) Plasma lipid concentration curve of the liposomes at predetermined times. (**B**) The area under the curve (AUC) of the lipid concentration of the liposomes in plasma at predetermined times. The bars represent mean ± SD (*n* = 3–4). Statistical significance was assessed by one-way ANOVA. * *p* < 0.05; *** *p* < 0.001; ns, no significance. (**C**) Timeline of IR783-sLip administration for in vivo imaging of tumor-bearing mice. (**D**) Representative in vivo dynamic fluorescence image. The dashed organe line indicates the tumor region. (**E**) Fluorescence intensity in the tumor ROI at the corresponding time points. (**F**) TBR at the corresponding time points. The bars represent mean ± SD (*n* = 3–4). Statistical significance was assessed by two-way ANOVA. ** *p* < 0.01; *** *p* < 0.001; ns, no significance.

**Figure 4 pharmaceutics-16-00744-f004:**
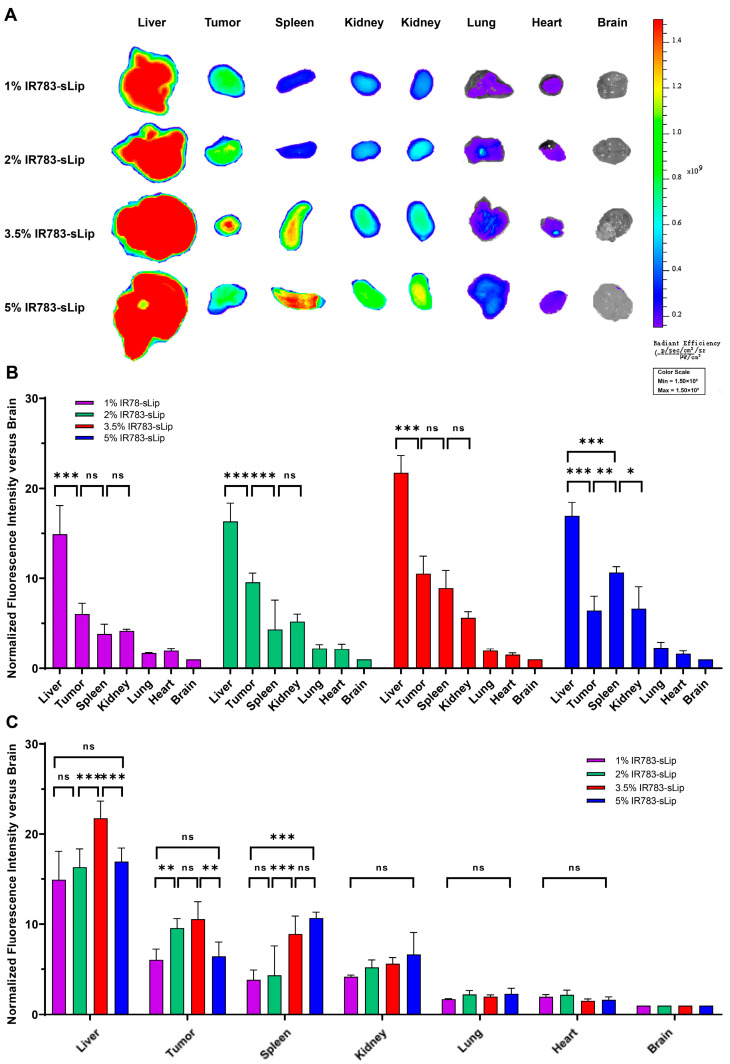
Effect of the density of conjugated IR783 on the biodistribution of IR783-sLip. Major organs of BALB/c mice intravenously dosed with 1%, 2%, 3.5%, and 5% IR783-sLip were dissected at the 24 h time point and imaged using NIR. (**A**) Representative ex vivo fluorescence image of major organs. (**B**) Normalized fluorescence intensity in major organs, compared within identical groups. (**C**) Normalized fluorescence intensity in major organs, compared in different groups. The bars represent mean ± SD (*n* = 3–4). Statistical significance was assessed by two-way ANOVA. * *p* < 0.05; ** *p* < 0.01; *** *p* < 0.001; ns, no significance.

**Figure 5 pharmaceutics-16-00744-f005:**
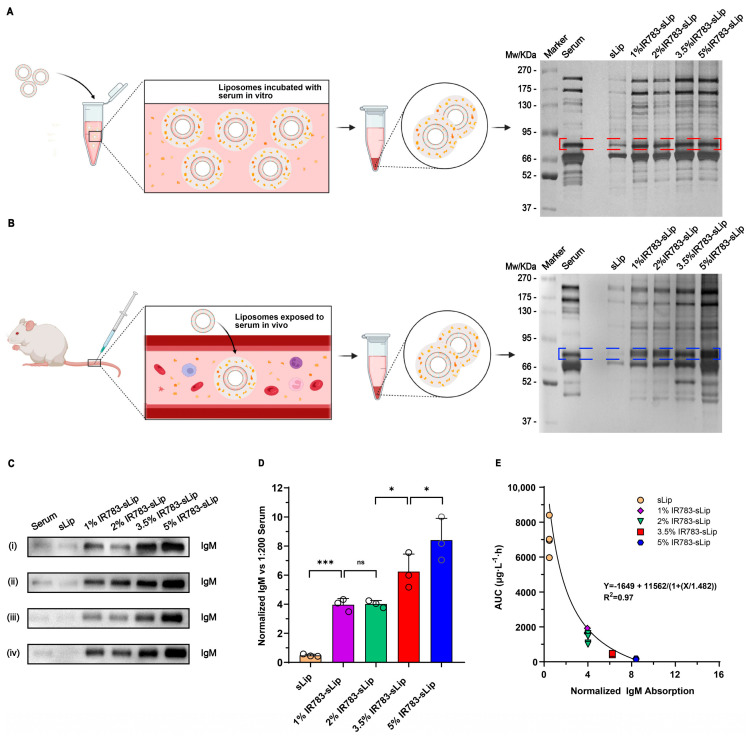
Effect of the density of conjugated IR783 on the liposomal absorption of the protein corona. (**A**) Silver staining of protein corona of liposomes formed in vitro and separated by SDS-PAGE. (**B**) Silver staining of protein corona of liposomes formed in vivo and separated by SDS-PAGE. (**C**) (i) Confirmation of dashed red line indicating protein IgM (mu chain at 72 kD) in the protein corona formed in vitro by Western blot. (ii–iv) Western blot of the dashed blue line indicating protein IgM (mu chain at 72 kD) in the protein corona formed in vivo. (**D**) Quantitative analysis, via Image J 1.35k software, of IgM in the protein corona formed in vivo and exhibited in (**C**) (ii–iv). The bars represent mean ± SD (*n* = 3–4). Statistical significance was assessed by two-way ANOVA. * *p* < 0.05; *** *p* < 0.001; ns, no significance. (**E**) Curve fit between the normalized IgM in the protein corona and the AUC data in vivo.

**Figure 6 pharmaceutics-16-00744-f006:**
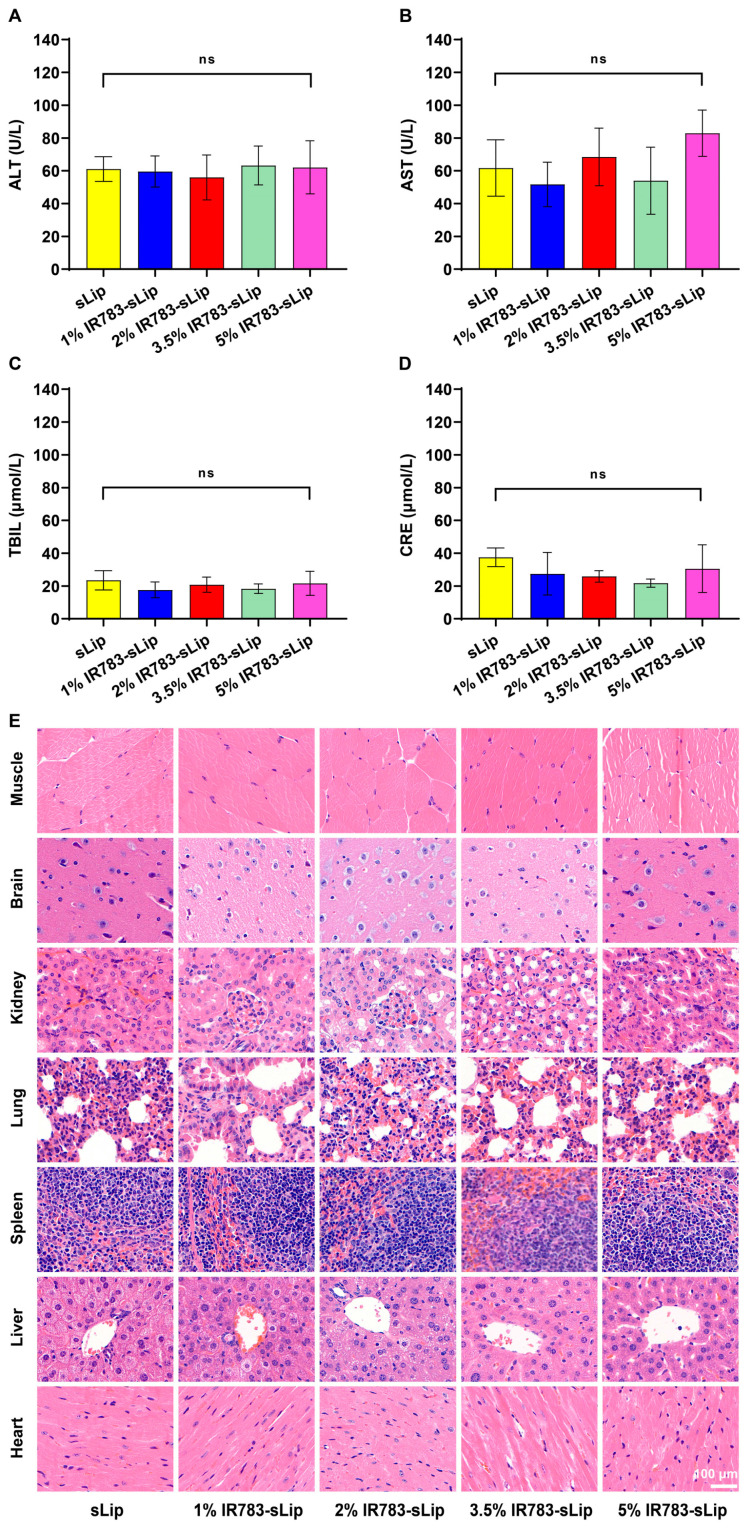
Safety evaluation of IR783-sLip in vivo. (**A**–**D**) Blood biochemical indexes (*n* = 3), represented by alanine transaminase (ALT), aspartate transaminase (AST), total bilirubin (TBIL), and creatine (CRE). (**E**) Observation of histological sections of major organs of mice receiving IR783-sLip once every other day for four treatments. The bars represent mean ± SD. Statistical significance was assessed by one-way ANOVA; ns, no significance.

## Data Availability

The data that support the findings of this study are available in the Appendix A of this article.

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
