# Peer review of "Effects of Surface IR783 Density on the In Vivo Behavior and Imaging Performance of Liposomes"

_pharmaceutics, 2024, doi:10.3390/pharmaceutics16060744_

Round 1

Reviewer 1 Report

Comments and Suggestions for Authors

The authors presented indeed very high-quality work on analyzing the influence of fluorescent dye density on the behavior of liposomes in vivo. Interestingly, it has been shown that more dye does not mean more efficient accumulation and there is a strictly defined optimum. I have virtually no comments on this work, and I believe that the work is worth publishing after minor adjustments:

Flow cytometry gating strategy should be shown and authors should explain the presence of the second peak in flow cytometry histograms. This second peak (if it is related to unwashed liposomes) can significantly alter the mean fluorescence intensity. 

Reviewer 2 Report

Comments and Suggestions for Authors

The authors have developed IR783-modified liposomes which were further evaluated for the cellular uptake efficiency, flow cytometry, and fluorescence microscope imaging. Interestingly, the effects of IR783 density on liposomal in vivo behavior were investigated by employing different studies.  In addition, the constitution of the protein corona was analyzed by Western blot assay. The topic is very interesting. The manuscript is well organized and covers all the crucial parameters to support the studies.

The manuscript can be accepted for publication.

I have just one query concerning zeta potential measurements:

The stealth and modified liposomes have shown positive zeta potential values. The authors should provide more information about the surface chemistry involved in it.

Good Luck

Reviewer 3 Report

Comments and Suggestions for Authors

This paper studied the effects of surface IR783 density on the imaging of liposomes. Through pharmacokinetic studies and imaging studies, they get the results of a correlation between IR783 density and protein corona absorption. But, the IR783 density does not improve the effectiveness of tumor imaging. So that optimizing the density of the modifications is recommended before using to the nanomaterial-based tumor imaging.

Comments:

1.      A549 cell is originally derived from a type II pneumocyte lung tumor, its liposomal uptake efficiency maybe different from other cell types. From the in vivo dynamic fluorescence image, we can see that liver cells may uptake liposomes more in animals. So that the results in A549 cells may not be used to analyze or compared with the phenomena in animals.

2.      The 5% IR783-sLip in vivo seems like the signal is saturated, so that it is hard to tell the tissue or organ difference in the animals.

Comments on the Quality of English Language

Need to be revised.

Round 2

Reviewer 3 Report

Comments and Suggestions for Authors

The authors answered all the questions and are resonable. The paper can be published in this version. 

Comments on the Quality of English Language

English can be improved more.